# IBCircuit: Towards Holistic Circuit Discovery with Information Bottleneck

## Abstract

Circuit discovery has recently attracted attention as a potential research direction to explain the nontrivial behaviors of language model (LM). It aims to find the computational subgraph, also known as *circuit*, that explains LM's behavior on specific tasks. Most studies determine the circuit for a task by performing causal interventions independently on each component. However, they ignored the holistic nature of the circuit, which is an interconnected system of components rather than an independent combination. Additionally, existing methods require redesigning a unique corrupted activation for each task, which are complicated and inefficient. In this work, we propose a novel circuit discovery approach based on the principle of Information Bottleneck, called IBCircuit, to identify the most informative circuit from a holistic perspective. Furthermore, IBcircuit can be applied to any given task without corrupted activation construction. Our experiments demonstrate the ability of IBCircuit to identify the most informative circuit in the model. The results from IBCircuit suggest that **the earlier layers in Transformer-based models are crucial in capturing factual information**.

## 1 Introduction

Circuits discovery in transformer-based language models usually involves identifying the subgraph (circuits) within the model which are responsible for solving specific tasks. Previous efforts to identify circuits within language models have led to the discovery of networks comprising attention heads and multi-layer perceptrons (MLPs) that either partially or fully explain the model's behaviors on tasks like indirect object recognition, modular arithmetic, docstring completion, and forecasting subsequent dates(Wang et al., 2022; Nanda et al., 2023; Hanna et al., 2024). However, the challenge is that transformer-based language models primarily operate as complicated black boxes when engaging in multi-layered nonlinear interactions within high-dimensional spaces (Wang et al., 2022), making it exceptionally difficult to understand their behavior.

Work in circuit analysis seeks to decode these models by reverse engineering (Conmy et al., 2023; Wang et al., 2022; Meng et al., 2022; Geva et al., 2021). Recent circuit analysis methods (Räuker et al., 2023), such as activation patching (Meng et al., 2022; Goldowsky-Dill et al., 2023) or knockouts (Wang et al., 2022), ignores the holistic nature of the circuit, which is an interconnected system of components rather than an independent combination. Additionally they scales poorly with the model size. Moreover, when faced with a task that has never been encountered before, most works need to redesign new corrupted activation for patching. Although recent works have proposed attribution patching (Syed et al., 2023) to estimate the importance of each edge in the computational subgraph without independent patching, they still need to design new corrupted activation for different task, which is inconvenient and complicated.

The Information Bottleneck (IB) leverages Shannon mutual information to quantify the compressed and informative nature of data distributions (Yu et al., 2022). This concept, rooted in information theory, aims to balance the trade-off between the complexity of the representation and its predictive power. The primary objective of IB is to distill a compressed yet predictive representation of the input signal (Tishby et al., 2000). By focusing on the most relevant aspects of the data, IB helps in reducing redundancy and noise, thereby enhancing the efficiency of data processing. This technique has been successfully applied in various domains, demonstrating its versatility and effectiveness. In feature selection, IB helps in identifying the most relevant features that contribute to the predictive

power of a model, thereby improving performance and reducing computational costs (Achille & Soatto, 2018b; Kim et al., 2021; Schulz et al., 2020). In representation learning, IB aids in creating compact and meaningful representations of data, which are crucial for tasks such as clustering, classification, and anomaly detection (Luo et al., 2019; Qian et al., 2020; Wu et al., 2020). These applications highlight the broad utility of IB in enhancing model interpretability and performance.

In this work, we propose the Information Bottleneck Circuit (IBCircuit), a novel approach designed to identify critical components within a Transformer-based model that are capable of task execution. The IBCircuit consists of two primary stages: **Model Perturbation** and **Component Selection**. During the **Model Perturbation** stage, the IBCircuit injects Gaussian noise into various components of the model, such as the activations of attention heads and MLP layers. Additionally, the *IB weights* are learned for the components to control the amount of noise added. The rationale for this noise injection is that it modulates the information flow from the original pretrained model to the perturbed version, with more significant noise leading to greater information distortion. As such, training the IBCircuit encourages the perturbed model to maintain its informativeness, which implies less noise injection. This process effectively approximates the condition of information compression, with less noise injection serving to identify the most informative components. In the **Component Selection** stage, the circuit is effectively formed by retaining these informative components. The contributions of this work are summarized as follows:

- We propose a novel circuit discovery method, IBCircuit, which utilizes information bottleneck to globally identify the most informative circuit of the model for circuit discovery from a holistic perspective.

- We introduce a model perturbation method that incorporates noise injection and adaptive learning of *IB weights*, which can be applied to any given task without corrupted activation construction.

- We validate the effectiveness of the IBCircuit in identifying the most informative attention heads in the Transformer-based model on the Indirect Object Identification (IOI) and Greater-Than Circuit Discovery task. The results from IBCircuit suggest that **the earlier layers in Transformer-based models are crucial in capturing factual information**.

## 2 RELATED WORK

**Circuit Analysis.** Circuit analysis seeks to understand machine learning models by identifying *circuits* in models that is responsible for given behaviors (Geiger et al., 2021; Wang et al., 2022; Conmy et al., 2023). Most of existing works conduct circuit analysis for language model leveraging activation patching (Zhang & Nanda, 2023) or its variants. Some overwrite activation values with zeros (Cammarata et al., 2021; Olsson et al., 2022), while others erase activation using the mean activation on the dataset (Wang et al., 2022; Hanna et al., 2024). Other works (Geiger et al., 2021; Wu et al., 2024) use the interchange interventions instead, which replaces the activation value of a node on one data point with its value on another data point. However, it has been justified that both zero and mean activations take the model too far away from actually possible activation distributions (Chan et al., 2022). Additionally, methods based on activation patching are inefficient as they require sequential operations on individual activations to assess their impact on task performance.

**Information Bottleneck in Deep Learning.** The principle of the information bottleneck (IB) aims to extract a compressed yet predictive code from the input signal (Tishby et al., 2000). Alemi et al. (Alemi et al., 2016) initially introduced the concept of the variational information bottleneck (VIB) to enhance deep learning. Currently, IB and VIB find applications primarily in representation learning and feature selection in deep learning. In representation learning, researchers aim to learn a compressed representation with the information bottleneck principle (Luo et al., 2019; Goyal et al., 2019; Qian et al., 2020; Wu et al., 2020). For feature selection, IB is used to select a subset of input features such as pixels in images or dimensions in vectors (Achille & Soatto, 2018b; Kim et al., 2021; Schulz et al., 2020; Yu et al., 2020), which are maximally predictive to the label of input data. Unlike previous work, we consider a rarely explored perspective: the compressed and relevant information in model components for specific behavior.

## 3 BACKGROUNDS

### 3.1 NEURAL CIRCUITS

Circuit discovery seeks to reverse-engineer model behavior by localizing it to subgraphs of the model's computation graph and explaining it. This approach aims to understand how specific parts of a model contribute to its overall functionality. Much research considers models as connected directed computational graphs, denoted as $G$. These graphs represent the flow of computations within the model, providing a structured way to analyze and interpret its operations. A transformer language model's (LM) computational graph is a directed graph (digraph) that describes the computations it performs. This graph flows from the LM's inputs to the unembedding layer, which projects its activations into vocabulary space. The nodes in this digraph are defined to be the LM's attention heads and multi-layer perceptrons (MLPs), though other levels of granularity, such as individual neurons, are also possible. This hierarchical structure allows for detailed analysis at various levels of abstraction.

A circuit is a subgraph that connects the inputs to the logits, which are the final outputs before the softmax layer in a language model. In this graph, source nodes represent the model's input, sink nodes represent the model's output, and intermediate nodes represent units of computation. By identifying and analyzing these circuits, researchers can pinpoint which parts of the model are responsible for specific behaviors and how information flows through the network. The concept of *Neural Circuits C* refers to induced subgraphs of $G$ that are responsible for specific behaviors and exhibit distinct functionality. These circuits can be thought of as the building blocks of the model's decision-making process. For instance, in Wang et al. (2022), the authors discover an Indirect Object Identification (IOI) circuit in GPT-2 small based on Activation Patching Meng et al. (2022).

### 3.2 TRANSFORMER ARCHITECTURE

A transformer model $G : \mathcal{X} \to \mathcal{Y}$ maps a token sequence $x = [x_1, \cdots, x_T] \in \mathcal{X}$ to a probability distribution $y \in \mathcal{Y}$. The $i$-th token at layer $l$ is embedded as a series of hidden state vectors $h_i^{(l)} \in \mathbb{R}^H$, where $H$ is the dimension of hidden state vectors. The input $h_i^{(0)}$ to the transformer is a sum of position $\text{pos}(i)$ and token embeddings $\text{emb}(x_i)$.

The internal computation of hidden states $h_i^{(l)}$ in $G$ can be summarized as follows: for each layer, it combines global attention $a_i^{(l)}$ and local MLP $m_i^{(l)}$ contributions computed from previous layers. Additionally, the *residual stream* draws information from previous states. The internal computations are as follows:

$$h_i^{(l)} = h_i^{(l-1)} + a_i^{(l)} + m_i^{(l)}, \tag{1}$$

$$a_i^{(l)} = \text{attn}^{(l)}\left(h_1^{(l-1)}, h_2^{(l-1)}, \ldots, h_i^{(l-1)}\right), \tag{2}$$

$$m_i^{(l)} = W_{proj}^{(l)} \sigma\left(W_{fc}^{(l)} \gamma\left(a_i^{(l)} + h_i^{(l-1)}\right)\right). \tag{3}$$

Each layer's MLP is a two-layer neural network parameterized by matrices $W_{proj}^{(l)}$ and $W_{fc}^{(l)}$, with rectifying nonlinearity $\sigma$ and normalizing nonlinearity $\gamma$. For each individual attention head, it is parameterized by four matrices $W_Q, W_K, W_V$ and $W_O$. The $QK$ matrix is used to compute the attention matrix $A = \text{softmax}\left(\frac{QK^T}{\sqrt{d_k}}\right) \in \mathbb{R}^{T \times T}$, where $d_k$ is the dimension of queries $Q$ and keys $K$, while the OV matrix determines what is written into the residual stream. For further background on transformers, we refer to Vaswani et al. (2017).

### 3.3 INFORMATION BOTTLENECK

The Information Bottleneck (IB) method is a powerful framework in information theory that focuses on finding a compressed representation of data while retaining the most relevant information for a specific task. The IB method aims to balance the trade-off between compressing the input data and preserving the information necessary for predicting an output variable. In essence, the Information Bottleneck method seeks to transform the input data into a new, more compact representation that still contains the critical features needed for accurate prediction or classification. This is particularly

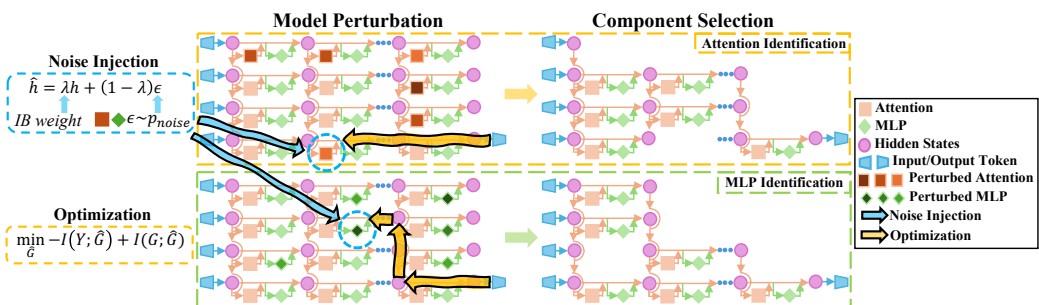

Figure 1: The framework of IBCircuit. During the model perturbation phase, we first inject noise to activation by using the *IB weight* $\lambda$, and then optimize $\lambda$ based on the Information Bottleneck objective. We represent attention components with more noise injected using darker orange squares, and MLP components with more noise injected using darker green rhombuses. The final selection is based on choosing components with the least amount of noise.

useful in scenarios where the original data is high-dimensional or contains a lot of noise, making it challenging to process directly. We denote $I(X;Y)$ the mutual information between two random variables X and Y, it is defined as follows:

$$I(X;Y) = \int_X \int_Y p(x,y)\log\frac{p(x,y)}{p(x)p(y)}\mathrm{d}x\mathrm{d}y. \tag{4}$$

Given the variable $X$ and its associated label $Y$, the Information Bottleneck (IB) Tishby et al. (2000) aims to learn the minimal sufficient variable $Z$ by optimizing the following objective:

$$\min_Z -I(Y;Z) + \alpha I(X;Z). \tag{5}$$

Here, $\alpha$ is the Lagrange multiplier used to balance the two terms. The first term encourages $Z$ to contain informative content about the label $Y$, while the second term minimizes the mutual information between $X$ and $Z$, ensuring that $Z$ only receives limited information from variable $X$.

## 4 METHODOLOGY

In this section, we introduce the proposed IBCircuit. In Section 4.1, we present the objective of IBCircuit in the Information Bottleneck framework. We outline the process of implementing Model Perturbation through Noise Injection and IBCircuit Optimization in Section 4.2. Additionally, in Section 4.3, we discuss the applications of Component Selection in Transformer-based models.

### 4.1 INFORMATION BOTTLENECK CIRCUIT

Circuit Identification aims to identify the most crucial components within a model, striking a balance between minimizing the number of components and maximizing their ability to perform tasks, which aligns perfectly with the concept of the Information Bottleneck (IB). Inspired by the IB, IBCircuit integrates circuit analysis into the information bottleneck framework to identify key components of pretrained models. Specifically, we denote $G$ as the set of all components of the pretrained model, $Y$ as the output of the pretrained model on a specific task, and $C$ as the circuit composed of critical components. We reformulate Equation equation 5 to obtain the objective of IBCircuit as follows:

$$\min_C -I(Y;C) + \alpha I(G;C). \tag{6}$$

This objective cannot be directly optimized since the mutual information is intractable to compute. Therefore, we propose an alternative objective to evaluate the compression quality of circuit $C$ by injecting noise into the intermediate activations of the pretrained model. As shown in Fig. 1, the IBCircuit consists of the Model Perturbation and Component Selection.

### 4.2 MODEL PERTURBATION

Our approach involves adaptively learning noise and injecting it into each activation, allowing the model to maintain similar performance on a specific task even after perturbation.

### 4.2.1 NOISE INJECTION

We define a pretrained model $G$ as a composition of $n$ components, denoted as $G = [v_1, v_2, \cdots, v_n]$, with the corresponding intermediate activations $\mathbf{h} = [h_1, h_2, \cdots, h_n]$. To introduce perturbations into $G$, we inject noise into intermediate activations with learnable *IB weights* $\lambda = [\lambda_1, \lambda_2, \cdots, \lambda_n]$. We sample noise $\epsilon$ from a parametric noise distribution. We insert Gaussian noise to the intermediate activations. *In high-dimensional space, random vectors tend to concentrate on a spherical surface, and their distribution can be approximated by a Gaussian distribution.* For each component $v_i$, we perturb the intermediate activation $h_i$ by combining it with noise $\epsilon$ using the *IB weight* $\lambda_i \in (0, 1)$:

$$\hat{h}_i = \lambda_i h_i + (1 - \lambda_i)\epsilon. \tag{7}$$

Here, $\hat{h}_i$ represents the perturbed activation. **In order to avoid introducing the noise that deviation from the original distribution, we calculate the mean and variance of the whole dataset as the variance of our injected noise.** The learnable *IB weight* $\lambda_i$ acts as a transmission probability, controlling the amount of information extracted from $h_i$ to $\hat{h}_i$. To ensure that $\lambda_i \in (0, 1)$, we define $\lambda_i = \text{Sigmoid}(\omega_i)$, where $\omega_i \sim \mathcal{N}(0, 1)$ is the learnable parameter. When $\lambda_i \to 1$, all the information from $h_i$ is transferred to $\hat{h}_i$ without loss. Conversely, when $\lambda_i \to 0$, $\hat{h}_i$ contains no information from $h_i$ but only noise. Unlike baseline methods that iteratively perturb each intermediate activation, the proposed method allows for simultaneous adjustment of the information flow from all activations $\mathbf{h}$ to $\hat{\mathbf{h}} = [\hat{h}_1, \hat{h}_2, \cdots, \hat{h}_n]$ by learning and updating all the *IB weights* $\lambda$ together. We denote the perturbed model as $\hat{G} = [\hat{v}_1, \hat{v}_2, \cdots, \hat{v}_n]$, we can then select the informative components of $\hat{G}$ into $C$ with learned *IB weights*. The selection process will be elaborated in Section 4.3.

### 4.2.2 OPTIMIZATION OF INFORMATION BOTTLENECK CIRCUIT

The perturbed model $\hat{G}$ is learned by extracting information from the pretrained model $G$ to achieve the same performance $Y$ on a specific task. On the one hand, we compress the effective information in $G$ by injecting noise. On the other hand, we aim to maximize the information content of the perturbed model $\hat{G}$ to achieve the performance $Y$. Therefore, the reconstructed IBCircuit objective is as follows:

$$\min_{\hat{G}} -I(Y; \hat{G}) + \alpha I(G; \hat{G}). \tag{8}$$

The first term encourages $\hat{G}$ to be sufficient for predicting $Y$, and the second term constrains the information that $\hat{G}$ learns from $G$. These two terms require us to inject noise into $G$ selectively so that $\hat{G}$ maintains valuable information as much as possible. The intuition is that injecting noises into the critical components of $G$ is more harmful to the functionality of $G$ than that into the irrelevant components. In that sense, the critical components are less likely to be injected with noise. Therefore, we can select the circuit $C$ from $\hat{G}$ by this criterion after training the IBCircuit.

We justify the above formulation through the following derivation. Let $G_s$ be a subset of $G$, which is independent to $Y$. Denote $G_\epsilon$ as the noisy subset of $G$ determined by injected noise, if we select the circuit $C$ by dropping $G_\epsilon$ in $G$, the following inequality holds:

$$I(G_s; C) \leq I(G_s; \hat{G}) \leq I(G; \hat{G}) - I(Y; \hat{G}). \tag{9}$$

This equation indicates when setting $\alpha = 1$ in Eq. equation 8, the IBCircuit objective upper bounds the mutual information of $G_s$ and $C$. Hence, optimizing the IBCircuit objective encourages $C$ to be less related to components in $G_s$ which are irrelevant to $Y$. The detailed proof is provided in Appendix A.

**Minimizing $-I(Y; \hat{G})$.** We first examine the first term $-I(Y; \hat{G})$ in Eq. equation 8, which encourages $\hat{G}$ to be informative of the output $Y$. We derive the upper bound of $-I(Y; \hat{G})$ as follows:

$$\begin{aligned} -I(Y; \hat{G}) &= \mathbb{E}_Y[\log p(Y)] - \mathbb{E}_{Y, \hat{G}}[\log p(Y|\hat{G})] \\ &\leq -\mathbb{E}_{Y, \hat{G}}[\log p(Y|\hat{G})] \\ &:= \mathcal{L}_{CE}(q_\theta(Y|\hat{G})) \end{aligned} \tag{10}$$

where $q_\theta(Y|\hat{G})$ is the variational approximation to the true posterior $p(Y|\hat{G})$. This inequality demonstrates that the minimization of $-I(Y; \hat{G})$ can be achieved by minimizing the training loss of

the model. Since we validate IBCircuit on the Transformer-based language model in this paper, we use the Cross Entropy Loss for the next token prediction training, denoted as $\mathcal{L}_{CE}$.

**Minimizing $I(G; \hat{G})$.** For the second term $I(G; \hat{G})$ in Eq. equation 8, which aims to extract the informative components from $G$ that contains minimal information about $G$. We minimize it by training the IBCircuit. By injecting more noise into insignificant components, while injecting less noise into more informative ones. Following Yu et al. (2022), by choosing the distribution of noise $\epsilon \sim \mathcal{N}(\mu_h, \sigma_h^2)$, where $\mu_h, \sigma_h^2$ are mean and variance of intermediate activations $\mathbf{h}$ in $G$. $I(G; \hat{G})$ has a tractable variational upper bound as follows:

$$I(G; \hat{G}) \leq \mathbb{E}_G(-\frac{1}{2}\log A_G + \frac{1}{2n}A_G + \frac{1}{2n}B_G^2) =: \mathcal{L}_{MI}(G; \hat{G}), \tag{11}$$

where $A_G = \sum_{i=1}^{n}(1 - \lambda_i)^2$ and $B_G = \frac{\sum_{i=1}^{n} \lambda_i(h_i - \mu_h)}{\sigma_h}$.

**Final Objectives.** Finally, the overall loss is defined as follows:

$$\mathcal{L} = \mathcal{L}_{CE} + \alpha\mathcal{L}_{MI}, \tag{12}$$

where $\alpha$ is hyperparameter used to adjust the weights of the loss.

### 4.3 COMPONENT SELECTION

We identify the critical components with less perturbation in the Component Selection stage.

**Attention Heads Selection.** To identify circuit attention heads, we perturb the attention matrix $A$ in each attention head of the Transformer module as $\hat{A}_i = \lambda_i A_i + (1 - \lambda_i)\epsilon$. We then define a threshold $\delta$ and select significant attention patterns between tokens based on the condition $\lambda_i > \delta$. Then we examine whether the corresponding attention patterns meet the definitions of critical attention heads to capture the model's circuit.

**MLP Layers Selection.** We perturb the hidden state m of each MLP layer defined in Eq. equation 3 to identify critical MLP layers, i.e., $\hat{m}_i^{(l)} = \lambda_i m_i^{(l)} + (1 - \lambda_i)\epsilon$. Similarly, we can find critical MLP layers by defining a threshold $\delta$ and identifying MLP layers where $\lambda_i > \delta$, or identify commonly occurring layers with larger $\lambda$ across multiple examples as critical MLP layersMeng et al. (2022).

## 5 EVALUATING IBCIRCUIT

To investigate the effectiveness of IBCircuit, we conduct the evaluation to answer the following research questions (RQs):

- **RQ1 (Grounded in Previous Work)**: Can IBCircuit effectively reproduce canonical circuits taken from previous works that found an end-to-end circuit explaining behavior for tasks?
- **RQ2 (Ablation Study)**: Are both CE loss and MI loss used for training IBCircuit necessary?
- **RQ3 (Equivalence)**: Does IBCircuit have the same chance as the pretrained model of outperforming each other?
- **RQ4 (Minimality)**: Does IBCircuit avoid including components which do not participate in the elicited behavior?

### 5.1 EXPERIMENT SETTING

**Tasks.** Although IBCircuit can be easily applied to LLMs like Llama, we primarily focus on GPT-2 small in this paper for better evaluation, as it is a model that is typically studied from a circuits perspective. We intentionally choose two tasks (IOI and Greater-Than) that have been studied before for easier comparison with previous work.

- **Indirect Object Identification (IOI)**: An IOI sentence involves an initial dependent clause, e.g., "When Mary and John went to the store", followed by a main clause, e.g., "John gave

a drink to Mary." In this case, the indirect object (IO) is "Mary" and the subject (S) is "John". The IOI task is to predict the final token in the sentence to be the IO. IOI Circuit Discovery aims to identify which attention heads of the model are crucial for performing such IOI tasks.

- **Greater-Than**: In the Greater-Than task, models receive input like "The war lasted from the year 1741 to the year 17", and must predict a valid two-digit end year, i.e. one that is greater than 41. In this paper, we aim to identify which attention heads of the model are crucial for predicting the end year.

**Baselines.** We compare the proposed method with the following methods:

- **Subnetwork Probing (SP)**. SP learns, via gradient descent, a mask for each node in the circuit to determine if it is part of the circuit or not, and encourages this mask to be sparse by adding a sparseness term to the loss function. The strength of this sparse penalty is controlled by a regularization hyperparameter.

- **Automated Circuit DisCovery (ACDC) (Conmy et al., 2023)**. ACDC traverses the transformer's computational graph in reverse topological order, iteratively assigning scores to edges and pruning them if their score falls below a certain threshold.

- **Attribution Patching (AP)**. AP assigns scores to all nodes at the same time by leveraging gradients information, and again prunes nodes below a certain threshold to form the final circuit.

We also compared two variants of IBCircuit, namely **IBCircuit-woMI** and **IBCircuit-woCE** in **RQ2**, which represent IBCircuit models trained solely with CE loss and MI loss, respectively.

**Metrics.** For the IOI task, we use logit difference (*logit diff*) for evaluation. Logit difference measures the difference in logits assigned to the correct and incorrect answers. For example, for the input "When Mary and John went to the store, John gave a drink to," we calculate logit(Mary)-logit(John). The larger the logit difference, the better the performance of the model or circuit. In the Greater-Than task, we use the *greaterthan* metric, which sums the total probability assigned to all correct and incorrect options and calculates the difference, e.g., for the input "The war lasted from the year 1741 to the year 17", we calculate $\sum_{y>41} P(y) - \sum_{y \leq 41} P(y)$. A larger difference indicates better model or circuit performance.

**Circuit Ablation.** We ablate the nodes that are not included in the circuit by using activation patching when evaluating the effectiveness of the identified circuit. We implement randomly selected activations from the corrupted dataset for patching. In the IOI task, we construct corrupted inputs by replacing IO and S with arbitrary names. In corrupted inputs of Greater-Than task, the start year's last two digits are changed to "01", leading models to output years prior to the start year.

## 5.2 RESULTS

**RQ1 (Grounded in Previous Work).** Following Conmy et al. (2023), we formulate circuit discovery as a binary classification problem, where nodes are classified as positive (in the canonical circuit taken from previous works) or negative (not in the canonical circuit). We determine a series of thresholds for ACDC, SP, AP, and IBCircuit by varying the number of nodes from 10% to 100%, increasing by 10% each time. We plot the pessimistic segments between the Pareto frontiers of TPR and FPR for each method across this range of thresholds.

Figure 2 illustrates the performance of IBCircuit in recovering the canonical circuit within GPT2-small, compared to existing methods. Our findings are as follows: i) IBCircuit shows competitive performance on both the IOI and Greater-Than tasks, notably outperforming baseline methods on the IOI task; ii) however, IBCircuit underperforms compared to ACDC on the Greater-Than task. This discrepancy may be due to the Greater-Than task input lacking a unique next token output label, unlike the IOI task. As a result, even with noise, the model can easily achieve the CE loss performance of the pretrained model, leading to inadequate training on the noisy model. The similar performance of IBCircuit-onlyMI and IBCircuit on the Greater-Than task further supports this observation.

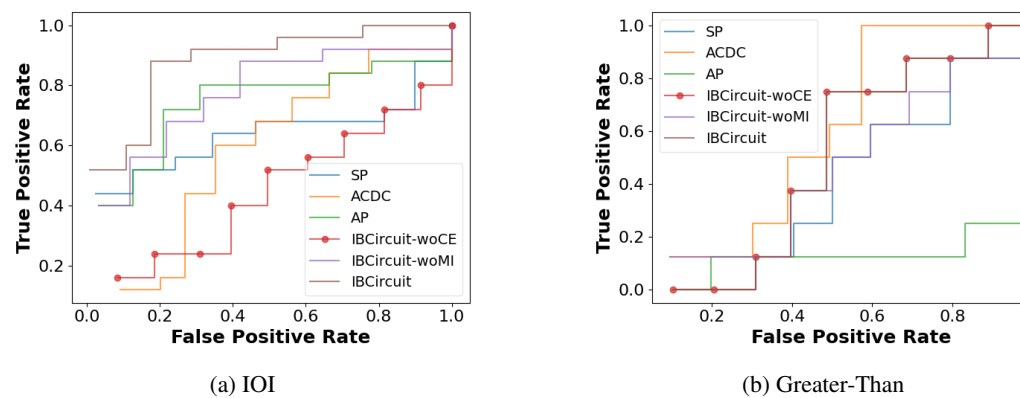

(a) IOI

(b) Greater-Than

Figure 2: ROC curves of SP, ACDC, AP and IBCircuit identifying model components from previous work, across IOI circuit and Greater-Than circuit.

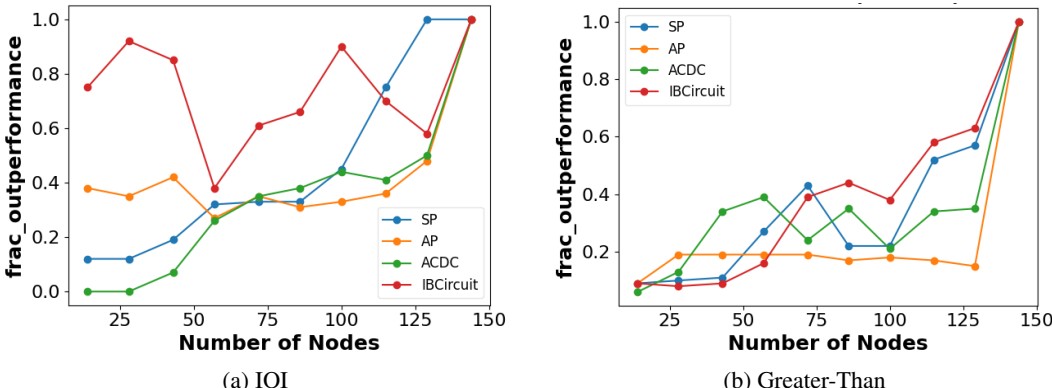

(a) IOI

(b) Greater-Than

Figure 3: Comparison of the equivalence of circuits found by different methods with the pretrained model. The higher the fraction of outperformance, the fewer the nodes, the better the circuit.

**RQ2 (Ablation Study).** In Figure 2, we compare the IBCircuit models trained without CE loss and without MI loss. We find that: i) on the IOI task, IBCircuit outperforms IBCircuit-woMI and significantly surpasses IBCircuit-woCE. This can be intuitively explained using Information Bottleneck, as the CE loss primarily serves to align the performance of the noisy model with that of the pretrained model, while the MI loss helps reduce irrelevant information in the noisy model. Consequently, the absence of MI loss in IBCircuit-woMI results in slightly worse performance compared to IBCircuit, whereas the lack of CE loss in IBCircuit-woCE severely diminishes the model's performance. ii) On the Greater-Than task, due to insufficient training from CE loss, IBCircuit and IBCircuit-woCE exhibit similar performance, both performing better than IBCircuit-woMI.

**RQ3 (Equivalence).** Intuitively, if the circuit can approximate the pretrained model, it should perform as well as the pretrained model. We assess the equivalence of the identified circuit with the pretrained model by calculating the proportion of instances where the circuit outperforms or performs equally to the pretrained model on both the IOI and Greater-Than tasks. For the IOI task, we calculate the proportion of instances with a higher or equal *logit diff*; while for the Greater-Than task, we calculate the proportion of instances with better or the same *greaterthan* metric.

Figure 3 shows the equivalence trends of circuits found by IBCircuit and related work based on different node number thresholds in comparison with the pretrained GPT2-small. In the IOI task, IBCircuit generally outperforms other methods across most node number thresholds. Specifically, IBCircuit demonstrates better performance than the pretrained model on over 50% of instances at most thresholds, especially when the node number is lower. In the Greater-Than task, for smaller node number thresholds, none of the methods yield circuits that perform better than the pretrained

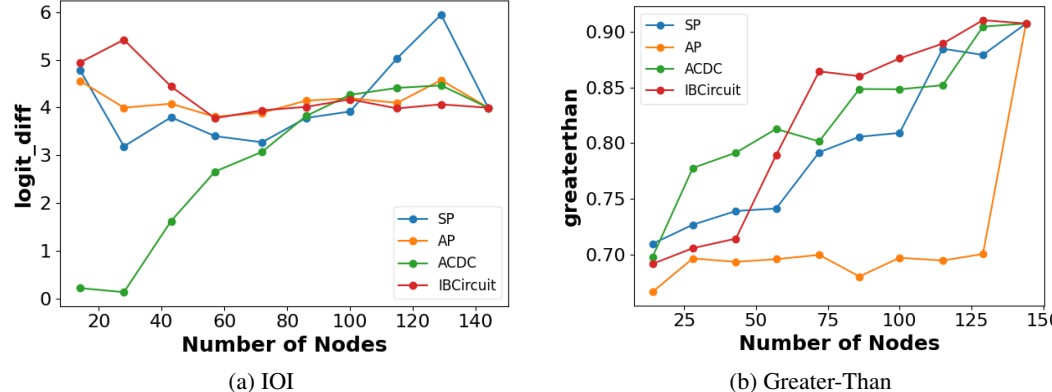

(a) IOI

(b) Greater-Than

Figure 4: Comparison of IBCircuit and related methods in terms of *logit diff* and *greaterthan* metrics under different node number thresholds. Higher metric scores and fewer nodes correspond to better circuits.

model on more than 20% of instances. However, at larger node number thresholds, IBCircuit surpasses other methods, achieving over 40% of instances performing better than the pretrained model.

**RQ4 (Minimality).** Intuitively, a circuit with fewer nodes that still achieves high metrics is less likely to contain components that do not participate in the behavior (Conmy et al., 2023). We measure the performance of various methods in terms of *logit diff* and *greaterthan* under different node number thresholds.

Figure 4 presents a detailed comparison of metric scores for various methods across different nodes in the Indirect Object Identification (IOI) and Greater-Than tasks. This figure provides a visual representation of how different approaches perform under varying conditions, offering insights into their relative effectiveness. By referring to Section 4.2 of the ACDC framework, we can conduct a thorough analysis of these results. In the IOI task, the IBCircuit method demonstrates superior performance compared to other methods. This is particularly evident as the number of nodes decreases, which results in a higher logit-diff. The higher logit-diff achieved by IBCircuit suggests that it is more effective at maintaining predictive accuracy even when the model is simplified by reducing the number of nodes. In the Greater-Than task, IBCircuit also outperforms other methods, especially when the greater-than probability exceeds 0.85. This threshold indicates a high level of confidence in the model's predictions, and the superior performance of IBCircuit in this range highlights its robustness and reliability.

## 6 CONCLUSION AND LIMITATIONS

In this paper, we aim to address the challenge of understanding the behavior of Transformer-based models, which are often seen as black boxes due to their complex computations. Traditional methods perform causal interventions independently on each component and require redesigning a unique corrupted activation for each task, which are complicated and inefficient. To overcome these limitations, we propose the Information Bottleneck Circuit (IBCircuit), a novel approach that leverages the Information Bottleneck to identify critical components. By injecting noise into model components and learning IB weights, the IBCircuit can effectively identify informative components while preserving their informativeness. Our experimental results in practical applications such as Indirect Object Identification (IOI) Circuit Discovery and Factual Recall Localization tasks demonstrate the effectiveness of IBCircuit in identifying informative attention heads and MLP layers.

The limitation of our approach is that the critical components identified by IBCircuit may vary across different tasks. Although we demonstrate its effectiveness in IOI Circuit Discovery and Greater-Than Localization, further investigation on other tasks is worth exploring. In future work, we aim to explore the applicability of IBCircuit in identifying critical components across other tasks, such as image-conditioned text generation Palit et al. (2023).

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

# A    PROOF OF IBCIRCUIT OBJECTIVE

The intuition is that injecting noises into the components of $G$ is more harmful to the functionality of $G$ than that into the irrelevant components. In that sense, the critical components are less likely to be injected with noise. Therefore, we can select the circuit $C$ from $\hat{G}$ by this criterion after training the IBCircuit.

We justify the above formulation through the following derivation. Let $G_s$ be a subset of $G$, which is independent to $Y$. Denote $G_\epsilon$ as the noisy subset of $G$ determined by injected noise, if we select the circuit $C$ by dropping $G_\epsilon$ in $G$, the following inequality holds:

$$I(G_s; C) \leq I(G_s; \hat{G}) \leq I(G; \hat{G}) - I(Y; \hat{G}). \tag{13}$$

This equation indicates when setting $\alpha = 1$ in Eq. equation 8, the IBCircuit objective upper bounds the mutual information of $G_s$ and $C$. Hence, optimizing the IBCircuit objective encourages $C$ to be less related to components in $G_s$ which are irrelevant to $Y$.

*Proof.* We follow the proof in Yu et al. (2022). Suppose $G$, $C$, $G_s$ and $Y$ satisfy the Markov condition $(Y, G_s) \rightarrow G \rightarrow C$ Achille & Soatto (2018a). Then we have the following inequality:

$$I(C; G) \geq I(C; Y, G_s) = I(C; G_s) + I(C; Y | G_s). \tag{14}$$

Since $Y$ and $G_s$ are independent, we have $H(Y|G_s) = H(Y)$ and $H(Y|G_s, C) \leq H(Y|C)$. Then we have:

$$I(C; Y | G_s) = H(Y|G_s) - H(Y|G_s, C) \geq H(Y) - H(Y|C) = I(C; Y) \tag{15}$$

Combine Eq. equation 14 and Eq. equation 15, we:

$$I(C; G_s) \leq I(C; G) - I(C; Y) \tag{16}$$

Suppose $G$, $\hat{G}$, $G_s$ and $Y$ satisfy the Markov condition $(Y, G_s) \rightarrow G \rightarrow \hat{G}$ Achille & Soatto (2018a). Then, combine with Eq. equation 16 we have:

$$I(\hat{G}; G_s) \leq I(G_s; G) - I(\hat{G}; Y) \tag{17}$$

$\hat{G}$ is deterministic given $G_\epsilon$ and $C$, since we can recover $\hat{G}$ by combining $G_\epsilon$ with $C$. Then for the left part in Eq. equation 17, we have:

$$I(\hat{G}; G_s) = I(G_\epsilon, C; G_s) = I(C; Gs) + I(C; G_\epsilon | G_s) \geq I(C; G_s) \tag{18}$$

Therefore, by combining Eq. equation 17 and Eq. equation 18 we have the follow inequality:

$$I(C; G_s) \leq I(\hat{G}; G_s) \leq I(\hat{G}; G) - I(\hat{G}; Y) \tag{19}$$

which proofs Eq. equation 13.

