# OpenReview forum: "IBCircuit: Towards Holistic Circuit Discovery with Information Bottleneck"
_ICLR.cc/2025/Conference — ICLR 2025 Conference Withdrawn Submission_

### Official Review · Reviewer_3QrS · 2024-10-28

**Soundness:** 2
**Presentation:** 2
**Contribution:** 2
**Rating:** 3
**Confidence:** 4

**Summary:**

This paper introduces a new method for node-level circuit-finding, IBCircuit. IBCircuit draws on the information bottleneck theory to find those components that are most responsible for the model's ability to perform a given task, without the need for corrupted examples. Fundamentally, it balances the model's ability to perform the task while ablating all non-circuit nodes with Gaussian noise, with the number of components in the circuit (or rather, the strength of the perturbation to each component). The authors then evaluate IBCircuit against existing methods (subnetwork probing, ACDC, and attribution patching) on two tasks (IOI and Greater-Than) using two existing metrics (ROC and model performance (logit/prob diff)) and one new metric (frac_outperformance). They claim that IBCircuit outperforms these other methods.

**Strengths:**

This paper has a nice theoretical motivation stemming from information bottleneck theory. Circuit analysis is an area of increasing interest, and this paper identifies some flaws (in particular with respect to components being scored independently) in current methods. The introduction of a new, demonstrably better circuit-finding method would definitely be an interesting paper.

**Weaknesses:**

- **Significant concerns with the proposed method**:
    - **Circuit analysis does not always aim to maximize task performance**: This paper claims (200-201) that "Circuit Identification aims to identify the most crucial components within a model, striking a balance between minimizing the number of components and maximizing their ability to perform tasks". This is false. The papers and methods it cites seek to do something (perhaps subtly) different: find all components (and edges) that are important to task performance, *whether their contribution is positive or negative*. See for example Wang et al.'s (2022) discovery of negative name mover heads, and Syed et al.'s (2023) use of *absolute* attribution scores. That is, we care about components that directly harm task performance as well. As the authors themselves state, IBCircuit is interested in maximizing model performance, and will likely miss these components.
    - **The role of counterfactual examples**: It is nice that this method requires no counterfactual examples, and the gaussian noise injection is carefully done so as to avoid out-of-distribution issues. However, I wonder whether this method isn't computing something different than what other circuit-finding methods find. They use counterfactual patching, using examples drawn from the same task; therefore, when they perform patching, they bake in the assumption that the model is performing a given task, and find only the circuit responsible for processing task examples. This method does not bake in that assumption, and will presumably also find components necessary for identifying the task as well. This is worth thinking about and discussing.
- **Large differences with other papers' conceptions of circuits**: The circuit papers that this paper compares to (Wang et al., 2023; Hanna et al. (2023)), as well as much of the related work (ACDC, edge attribution patching) conceive of the model's computational graph as a graph $G=(V,E)$, and a circuit as a subgraph of that. Notably, a graph includes both nodes *and edges*, but IBCircuit works only at the node level. I guess the authors used a node-level adaptation of attribution patching (and ACDC?) when evaluating, as is necessary for a fair comparison. But regardless of whether they did, IBCircuit, working on nodes alone, seems much less powerful than the other methods which can find edges as well.
- **Limited / flawed evaluation**:
    - **Flawed evaluation methodology**: The evaluations of IBCircuit are pretty flawed and mixed in results:
        - Due to the aforementioned node vs. edge issue, IBCircuit (a node based method) compares against edge-based methods, and it's not clear if they (especially ACDC) have been adapted into node-based variants. Moreover, the evaluation performed (node-based ablations) are thus different from how those original papers evaluated their results. We can see this discrepancy clearly in Figure 2(b), where attribution patching has very poor performance, while in the original paper that introduced it, the same curve (for the same task) looks much better, for both attribution patching and ACDC. I'm not sure if this is due to the authors' implementation of attribution patching, because of the node-based ablations, or something else, but something is wrong here.
        - The authors report logit and probability difference for IOI and greater-than, but don't report the base model's performance on each. It would be good to do so, because oftentimes circuit work aims to find a circuit that replicates the original model's performance; thus, as discussed above, higher performance isn't always better.
        - The authors invent a new metric (frac_outperformance) in Figure 3, but it's not clear why it's valuable. As discussed earlier, we don't really want the circuit to outperform the whole model. Indeed when ablation is performed using corrupted examples + patching, as this paper does, it's easy to get high performance at low node counts by ablating nodes (like negative name mover heads) that are important, as this will force those negative heads to behave as positive heads. But this isn't really desirable; it actually means you have missed many important components.
        -  IBCircuit's performance across Greater-Than tasks seems pretty middling, even by this paper's flawed metrics. Evaluating on other tasks (see [Hanna et al. (2024)](https://arxiv.org/abs/2403.17806) for other circuit-related tasks) would more convincingly prove IBCircuit's value.
    - **No discussion of or comparison to recent circuit-finding methods**: Beyond the circuit-finding methods (ACDC and attribution patching) currently discussed, there are many new ones that go unmentioned. [Bhaskar et al. (2024)](https://arxiv.org/abs/2406.16778v1), for example, use a learnable mask over model edges to find circuits, which also answers the objection about methods that consider nodes/edges independently; see similar ideas from [Chintam et al. (2023)](https://aclanthology.org/2023.blackboxnlp-1.29/). [Hanna et al. (2024)](https://arxiv.org/abs/2403.17806) and [Marks et al. (2024)](https://arxiv.org/abs/2403.19647) introduce new versions of (edge) attribution patching that improve its accuracy as well. While Bhaskar et al. is somewhat new (3 months old), the edge attribution papers have been out for half a year, and are definitely worth discussing, and even comparing against.
- **Unsupported claims about factual recall**: In the abstract (23-34), the authors claim "the results from IBCircuit suggest that the earlier layers in Transformer-based models are crucial in capturing factual information"; a similar claim appears in the conclusion. But no evidence of this appears in the text; no factual recall tasks are even studied.
- **Proofreading**: In general, the manuscript could use some proofreading; see the list of typos below.

Overall, this paper overlooks many theoretical concerns about circuit finding, and performs insufficient evaluation to prove that its proposed method is actually better / somehow more useful than prior methods.

**Questions:**

## Questions
- Is IBCircuit capable of finding edges between nodes?
- Is IBCircuit capable of finding nodes whose importance is negative (i.e. they are important but act against the model's task abilities)?
- Do you adapt ACDC to be just a node-based metric? I guess this would look similar to vanilla activation patching.
- Why do the ROC curves in this paper look different from the ones in Syed et al. (2023); is it all up to the node vs edge patching distinction?
## Comments/Typos
- (33-34): missing space before citation
- (39-41): "Recent circuit analysis methods (R ̈auker et al., 2023), such as activation patching (Meng et al., 2022; Goldowsky-Dill et al., 2023) or knockouts (Wang et al., 2022), **ignore**"
    - Relatedly, you should cite Vig et al. (2020) or Geiger et al. (2020) for activation patching, as these originated the method in the context of interpretability on LMs.
- (41): Additionally they **scale** poorly
- (43-46): This statement is false; Syed et al.'s method estimates edge importance independently for each edge as well.
- (93): Hanna et al. (2023) don't use mean ablation as you state.
- (94): drop "the" before "interchange interventions"
- (100-101): Use \citet to cite the paper like Alemi et al. (2016)
- (180): remove "it is"
- (205-206): repeated "Equation 5 equation 5" (here and elsewhere)
- (248-249): "injecting **noise**"
- (275-276): sentence fragment
- (291, 296): Attention **Head** / MLP **Layer** Selection
- (318): Cite Llama and GPT-2 here.
- (343): Where is the citation for attribution patching? There is one for ACDC. Same for subnetwork probing

---

### Official Review · Reviewer_HkKb · 2024-11-01

**Soundness:** 2
**Presentation:** 2
**Contribution:** 3
**Rating:** 3
**Confidence:** 4

**Summary:**

This paper introduces Information Bottleneck Circuit (IBCircuit), a framework to identify a circuit within a model responsible for a task. Inspired by information theory, IBCircuit consists of two steps, noise injection and component selection. The paper demonstrates the effectiveness of IBCircuit on two tasks, IOI and Greater-than, by showing this method can obtain circuits that recover manual circuits better, while maintaining comparable performance to the original model with minimal nodes.

**Strengths:**

- Leveraging information theory to address circuit discovery for mechanistic interpretability is original.
- To automate circuit discovery with scalability is an important problem. This work, with all the questions clarified and more rigorously vetted, could have potentially significant impact to the community if adopted widely as a standard technique.

**Weaknesses:**

- The authors should quantify the efficiency gain of IBCircuit as it is one of the main claims in the paper. For example, the authors should provide runtime comparison between different circuit discovery methods to support the claim.
- The paper is missing a highly relevant work: Sparse Autoencoders Enable Scalable and Reliable Circuit Identification in Language Models by O'Neill and Bui, where they use SAE to identify holistic circuits in transformers efficiently. Due to the high relevance, the authors should specifically compare the key technical differences between IBCircuit and the SAE approach, and discuss potential advantages or disadvantages of each method, and incorporate the prior work as one of the baselines.
- Since IBCircuit involves model training and hyperparameter tuning (i.e., the $\alpha$ in the objective function), the authors should conduct an ablation study showing how different values of $\alpha$ affect the performance and characteristics of the identified circuits.
- The paper didn't discuss the reproducibility of the method or plans to release. It's hard to evaluate the quality of their code for reproducibility purpose either since it's not provided as one of the supplementary files.
- Writing skill could be improved: a lot of grammatical/formatting errors found and the flow is not fluent. (See **minor issues** in the questions section). Readers might find certain parts hard to understand as a result.
- The paper is missing one of the standard tasks often evaluated in circuit discovery work, the docstring task. The authors should consider including the docstring task to be more comparable to prior work.
- In addition to comparisons between the identified circuits and the original models, the authors should also consider including another important effectiveness experiment: comparisons between the identified circuits with random circuits as random circuits are often a strong baseline.

**Questions:**

- How are the datasets partitioned? How much data is used for training and validation? Is the evaluation done on a separate test set?
- The authors claim that “the earlier layers in Transformer-based models are crucial in capturing factual information.”, but I don’t see any experimental results presented in the paper to support this claim. The authors should provide specific experimental evidence supporting this claim or to remove or qualify the statement if it's not directly supported by their results.
- Regrading the ACDC performance reported in the paper, is it on task-specific metrics or KL divergence? The authors should specify since it’s known that ACDC can achieve better results using KL divergence.
- What is the definition of “nodes” in the evaluated circuits? Since the max number of nodes in Figure 3 and Figure 4 is around 144, did the author only consider attention heads as nodes? If the IBCircuit framework and all the baselines are able to include both attention heads and MLPs in the identified circuits, why not include MLPs too in the circuits evaluated?
- Regrading the necessity of the RQ4-Minimality evaluation: Figure 3 and Figure 4 essentially show the same trend, meaning we can derive both RQ3 and RQ4 from only Figure 3, so Figure 4 seems to be redundant. Furthermore, I don't agree with this claim in the RQ4 discussion: "The higher logit-diff achieved by IBCircuit suggests that it is more effective ...". Since the goal of circuit discovery is to identify a circuit that emulate the original model performance, why obtaining a "higher" logit-diffis desirable? This could potentially mean the circuit fails to capture some negative contributing nodes, such as the negative name mover heads in the IOI task (which is also critical when we want to emulate the original model performance). For circuit discovery, it's more important to recover the original model performance (i.e., faithfulness = 1) instead of pursuing a high prediction accuracy/high logit-diff.
- The ROC AUC of AP on Greater-Than in Figure 2 seems to be unusually low compared to what's reported in prior work (Attribution Patching Outperforms Automated Circuit Discovery). Can the authors clarify how the AP baseline is run?
- Confusion about line 482-482. “The limitation of our approach is that the critical components identified by IBCircuit may vary across different tasks.” The composition of circuits should naturally differ for different tasks: why is this a limitation?

**Minor Issues**
- The authors should consider using consistent color to represent baseline methods in figures. For example, AP was represented using green in Figure 2 but orange in Figure 3.
- Line 40: The “knock-outs” used in Wang et al. is also activation patching. The current form of the sentence, “such as activation patching or knock-outs” might confuse the reader.
- Line 130, 299: should use \citep instead of \citet for Meng et al 2022.
- Line 183: should use \citep instead of \citet for Tishby et al 2000.
- Line 222: Unjustified claim for the use of Gaussian noise: “In high-dimensional space, random vectors tend to concentrate on a spherical surface, and their distribution can be approximated by a Gaussian distribution.” Please provide evidence for this claim, or add in citations if this has already been shown in prior work.
- Line 227-228: Unclear meaning of this sentence: “...we calculate the mean and variance of the whole dataset as the variance of our injected noise”. Did the author try to say as the “mean” and variance of our injected noise?
- Line 273: redundant word in “in Eq. equation 8”.
- Line 296: redundant word in “in Eq. equation 3”.
- Line 457: grammar error: “...for various methods across different nodes…” should be “for various methods across different number of nodes”
- Line 485: should use \citep instead of \citet for  Palit et al. (2023).
- Line 610, 620: should use \citep instead of \citet for Achille & Soatto (2018a).
- Line 627: the S in I(C;G_S) is not consistent with that in other terms in this equation.

---

### Official Review · Reviewer_vzZw · 2024-11-04

**Soundness:** 3
**Presentation:** 3
**Contribution:** 3
**Rating:** 5
**Confidence:** 4

**Summary:**

This paper presents a novel circuit discovery method utilising the Information Bottleneck analysis method. Their framework has two phases. First, they inject noise to model activations to establish the "importance" of each components. Then, in step 2, they identify the circuit by selecting the components with the minimum amount of noise injected. Their method achieved better results compared to ACDC and the baselines used by ACDC.

The paper made some claims about the inner-workings of the model that have not been elaborated. For example: "the earlier layers in Transformer-based models are crucial in capturing factual information" was highlighted in the abstract. This particular claim is not at all elaborated in the main content part of the paper.

**Strengths:**

The paper present a new circuit discovery method with better performance compared to ACDC and the baselines used by ACDC.

**Weaknesses:**

As mentioned in the "Summary" section, the claim about the internal organisation of the model is not substaintated in the main content section. The authors claimed that "the earlier layers in Transformer-based models are crucial in capturing factual information," but this is not discussed anywhere in the remainder of the paper.

Furthermore, this claim directly contradcits Meng et al.'s (2022) hypothesis, which has been cited by the authors. They claimed that factual information is located in the middle layers. Let alone the discussion on "BERT rediscovers the classical NLP pipeline" (Tenney et al., 2019) This layer view of the model is still a highly controversial topic.

**Questions:**

Please see the weakness section.

---

### Official Review · Reviewer_vrns · 2024-11-06

**Soundness:** 2
**Presentation:** 3
**Contribution:** 2
**Rating:** 3
**Confidence:** 4

**Summary:**

This work proposes a circuit discovery method based on the information bottleneck, aiming to identify the most informative circuit from a holistic perspective. The motivation behind this work is that existing methods primarily rely on independent interventions on components of the computational subgraph, often overlooking the interconnected relationships between components. The core contribution of this work is a model perturbation method that uses noise injection and adaptive learning of IB weights to identify critical components without needing corrupted activations. The effectiveness of the proposed method is validated on the IOI and Greater-Than tasks.

**Strengths:**

S1. This work applies the Information Bottleneck principle to the circuit discovery task, though the innovation is somewhat limited.

S2. The experimental results support some of the authors' claims.

**Weaknesses:**

W1. The innovation in this work is limited. The method of using Gaussian noise for perturbation has been applied in prior work, such as the referenced paper by Yu et al. (2022), particularly in sections 4.2.1 and 4.2.2.Although the authors mention being inspired by prior work, the core contribution is too similar to it.

W2. I believe that circuit discovery shares similarities with several existing tasks, including explainability in graph networks. Integrating key ideas from leading methods in related tasks into circuit discovery is valuable; however, I would have liked to see more substantial contributions beyond adapting these existing approaches.

W3. The paper claims the method can scale to larger models, but lacks experiments to support this.

**Questions:**

As noted in the limitations, the paper would benefit from additional experiments on a wider range of tasks.

---

### Note · Authors · 2024-11-30

**Comment:**

We thank all the reviewers for their valuable feedback. We plan to address the raised points for a revised version of the work and, as such, withdraw the current paper.

**Withdrawal Confirmation:**

I have read and agree with the venue's withdrawal policy on behalf of myself and my co-authors.